# The Effect of a Knee Brace on Muscle Forces during Single-Leg Landings at Two Heights

**DOI:** 10.3390/ijerph20054652

**Published:** 2023-03-06

**Authors:** Yubin Wang, Haibin Liu, Huidong Wei, Chenxiao Wu, Feijie Yuan

**Affiliations:** 1School of Sport and Health Science, Dalian University of Technology, Dalian 116024, China; 2College of General Aviation and Flight, Nanjing University of Aeronautics and Astronautics, Nanjing 213300, China

**Keywords:** single-leg landings, knee brace, musculoskeletal modeling, ACL injuries

## Abstract

Single-leg landing is one of the maneuvers that has been linked to non-contact anterior cruciate ligament (ACL) injuries, and wearing knee braces has been shown to reduce ACL injury incidence. The purpose of this study was to determine whether wearing a knee brace has an effect on muscle force during single-leg landings at two heights through musculoskeletal simulation. Eleven healthy male participants, some braced and some non-braced were recruited to perform single-leg landings at 30 cm and 45 cm. We recorded the trajectories and ground reaction forces (GRF) using an eight-camera motion capture system and a force platform. The captured data were imported into the generic musculoskeletal model (Gait2392) in OpenSim. Static optimization was used to calculate the muscle forces. The gluteus minimus, rectus femoris, vastus medialis, vastus lateralis, vastus medialis medial gastrocnemius, lateral gartrocnemius, and soleus muscle forces were all statistically significant different between the braced and non-braced participants. Simultaneously, increasing the landing height significantly affected the gluteus maximums, vastus medialis, and vastus intermedia muscle forces. Our findings imply that wearing a knee brace may alter muscle forces during single-leg landings, preventing ACL injuries. Additionally, research demonstrates that people should avoid landing from heights due to the increased risk of knee injuries.

## 1. Introduction

Jumping and landing are the most common maneuvers in sports and are prone to cause non-contact injuries such as anterior cruciate ligament (ACL) injuries [1,2,3]. Around 70% of ACL injuries occur as a result of landing or sidestep cutting tasks [4]. Single-leg landings and double-leg landings are the two types of landings. Due to lower knee flexion angles, single-leg landings are more prone to induce ACL injuries than double-leg landings are [5,6]. ACL injuries not only decrease an individual’s daily activity but also lead to higher risks of degenerative joint disease [7].

Muscle force is critical in preventing ACL injuries and maintaining knee stability. Regarding muscle force, weak quadriceps are a risk factor for non-contact ACL injuries [8]. Similarly, soleus and hamstring coordination may offer additional support in reducing the risk of ACL injuries during single-leg landings [9]. In addition, some investigations have found that the gastrocnemius and soleus muscles (which stabilize the ankle joint) can also prevent ACL injuries [10,11].

Reduced activation of the quadriceps may be beneficial in preventing ACL injury during single-leg landings [12]. The muscle activation ratio responses of the soleus and gastrocnrmius, as well as of the quadriceps and hamstring, should be considered when assessing ACL injury risk during landing [13]. Previous studies have shown that wearing a knee brace can influence the degree of activation of the lower limb muscles. For example, surface electromyography (EMG) signals show that the vastus lateralis and vastus medialis muscles are extensively active when wearing a knee brace, as was found in a survey conducted on kicking [14]. In another study on squatting, the muscle activation level represented as the peak root mean square average of the surface EMG signal declined by around one third [15]. It should be noted that observing surface EMG signals is an approach with limitations as the signals suffer crosstalk from surrounding, particularly deeper muscles. In addition, the relation between the EMG signal and muscle force varies across activation conditions (e.g., isometric/concentric/eccentric). For example, a linear relation is typically only held under isometric contraction. Hence, a surface EMG signal alone will not be able to offer us a complete and detailed blueprint of the activation of muscles involved. Compared to the employment of surface EMG signals to study muscle force, the simulation approach in this research can obtain more muscle force data of different muscle parts in a non-invasive manner. However, the effects of wearing a knee brace on muscle force during single-leg landings have never been studied by simulation.

Therefore, the purpose of the study was to investigate the effects of wearing a knee brace on lower limb muscle activation during single-leg landings at two heights by simulation. Specifically, we hypothesized that wearing a knee brace can affect lower limb muscle activation. Regarding landing height, we hypothesized that muscle forces would be increased with the increase in landing height.

## 2. Methods

### 2.1. Participants

Eleven physically active male participants were recruited to conduct landing tasks (the sample size was set according to experience). None of the participants had a history of lower limb injuries within at least six months before the study. All participants (age: 23.2 ± 2.5 years; height: 1.78 ± 0.34 m; mass: 74.2 ± 6.1 kg) were physically active according to the international standard of physical activity guideline, which defines physically active lifestyle as exercising for at least 150 min at a moderate intensity or 75 min at a vigorous intensity every week in three months [16]. Informed consent was provided, and the Biological and Medical Ethics Committee of the Dalian University of Technology approved this study.

### 2.2. Instrumentation

In this study, we used the Ossur elastic knee brace (~160 g net weight), which was authenticated by European Union certification and International Organization for Standardization certification. Two multifunctional boxes of the heights 30 cm and 45 cm were employed. For a motion analysis, an 8-camera motion capture system with a frequency of 100 Hz (Vicon Peak, Oxford Metrics Ltd., Oxford, UK) was used to record three-dimensional (3D) marker trajectories. Thirty-nine retro-reflective markers (25 mm diameter) were attached to each person at the corresponding anatomical landmarks based on the Gait2392 model [17]. In addition, a force platform (AMTI, Advanced Mechanical Technology, Inc., Watertown, MA, USA) was utilized synchronously to record ground reaction forces (GRF) at a frequency of 1000 Hz. Kinematics and kinetic data were filtered by a zero-phase lag and a fourth-order Butterworth filter with a cut-off frequency of 6 Hz.

### 2.3. Experiment Procedure

The duration of a single-leg landing was defined as the interval starting from initial foot contact to maximum knee flexion [18]. Before the test, the participants were told to complete a 5 min warm-up (jogging) upon arriving at the sports science lab. Then, details about the single-leg landing were explained to help them understand the form of the maneuver. The participants performed single-leg landing three times for each test condition using their dominant leg, defined as the limb that people prefer to kick a ball with [19]. All the participants used their right lower limbs to perform the task in this study. The experiments proceeded in four categories with two factors (height and knee brace). Each participant completed one trial, and a 30 s break was given to avoid fatigue. A successful test was accepted if the participant stepped off the box in a stable landing posture without an upward and forward jumping motion [9]. Injuries are more likely to occur at peak GRF and maximum knee flexion during landings [18,20].

### 2.4. OpenSim Simulation

OpenSim 4.0, an open-source musculoskeletal simulation software [21], was run to obtain the kinematics, kinetics, and muscle forces. First, C3D files obtained from Vicon were transferred by MATLAB (The Mathworks Inc., Natick, MA, USA) and were imported into the OpenSim. A musculoskeletal model named Gait2392 was established in the platform, which possesses eight segments (torso, pelvis, femurs, shanks, and feet), 23 degrees of freedom, and 92 muscle–tendon units, mainly in the lower limbs (Figure 1). A generic model of each participant was obtained through a running scale according to anthropological data captured in the lab [22,23,24]. Specifically, the dimension of each segment was scaled according to corresponding markers obtained from the static experiment captured in the lab, and the RMS error in the position of the markers was restricted to under 3 cm. Then, inverse kinematics was run to minimize the differences between the experimental and virtual mark positions through the least-squares method, and its RMS mark errors were limited to under 3.5 cm [25]. Afterward, joint angles were obtained, a primary outcome measure for conducting the following steps. Inverse dynamics was run to obtain the net joint torque using the results of the last step (i.e., inverse kinematics) and the GRF to solve a series of dynamic equations. In addition, a residual reduction algorithm (RRA) was adopted to reduce the residual force (applied to the pelvis) caused by the inconsistency between the force platform and the kinematics data in the musculoskeletal model. Finally, static optimization predicted muscle forces (using the RRA results) with the objective function being the minimization of the sum of the squared muscle activations [26].

### 2.5. Statistical Analysis

The gluteus maximus, gluteus medius, gluteus minimus, rectus femoris, medial femoris, lateral gastrocnemius, and soleus muscles were examined. The experimental data of the outcomes of this study were expressed as mean ± standard deviation (means ± S.D). The muscle force data were divided by each participant’s body mass via Excel 2018, and two characteristic moments that could both cause ACL injuries were selected based on previous literature reports: the moment at which the peak GRF was observed and the moment at which maximum knee flexion occurred. The retrieved data were stored and counted using the SPSS 22.0 (SPSS Inc., Chicago, IL, USA) software. A two-way repeated measures ANOVA was utilized to assess if wearing a knee brace alters muscle force during single-leg landings at two heights in a musculoskeletal simulation. A significance level of 0.05 was observed. The final graphics were drawn using Prism 8.0.

## 3. Results

Initially, we compared our results with those of a previous study (Figure 2) [13], and the similar patterns observed for the quadriceps, hamstrings, and gastrocnemius muscle activations indicate that the simulation results were reliable.

At the peak GRF, no correlation between whether or not a knee brace was worn and the landing height was indicated by any of the lower limb muscle forces (*p* > 0.05). The outcomes indicated significant differences (*p* < 0.05) in the forces of the vastus medialis, vastus intermedius, vastus lateralis, medial gastrocnemius, lateral gastrocnemius, and soleus in the participants who were wearing a knee brace compared to those who were not wearing one. The gluteus maximus muscle was significantly affected at the height of 45 cm compared to 30 cm.

At maximum knee flexion, no correlation between whether or not a knee brace was worn and the landing height was indicated by any of the lower limb muscle forces (*p* > 0.05). Significant differences in the forces of the gluteus minimus, vastus medialis, vastus lateralis, rectus femoris, vastus intermedius, medial gastrocnemius, and lateral gastrocnemius were observed both in the participants with a knee brace and in those without one (*p* < 0.05), with the muscle forces being larger in the participants who were wearing a knee brace, except for the muscle force of the medial gastrocnemius. Notably, the forces of the vastus medialis and vastus intermedius were also significantly larger when the participants landed from the 45 cm height compared to when the landed from the 30 cm height, with the vastus medialis showing the most significant increase of around 50%.

## 4. Discussion

This is the first study to examine the effect of wearing a knee brace on lower limb muscle forces during single-leg landings at two different heights using OpenSim simulations. The purpose of this study was to evaluate whether wearing a knee brace had an effect on the muscular forces generated during single-leg landings at two different landing heights. Our research revealed that wearing a knee brace may help avoid ACL injuries by strengthening the lower limb muscle. The results supported our first hypothesis and demonstrated the important role of wearing a knee brace during single-leg landings.

Our findings are not directly comparable to those of earlier studies due to the differences in methodology and maneuvers. For example, one study discovered a significant decrease in quadricep activity during the performance of cutting maneuvers while wearing a brace [27]. A significant increase in quadriceps force was found in the gait support phase [28]. However, these two experiments used surface EMG to assess muscle activity. In contrast, we simulated it using static optimization, which is a more efficient strategy [29]. Our findings suggest that wearing a knee brace increases quadricep force during single-leg landings at the maximal knee flexion angle. Additionally, we observed a significant increase in VAS (rectus, vastus medialis, and vastus lateralis) muscle forces during the performance of single-leg landings while wearing a brace, which is a similar finding to that of a simulation investigation of the effect of wearing a knee brace during double-leg landings [30]. Quadricep forces are critical to dynamic knee stability and avoiding ACL injuries [31,32]. Our findings imply that wearing a knee brace may help avoid ACL injuries by increasing quadricep muscle forces.

Additionally, wearing a knee brace had a significant influence on the gastrocnemius and soleus muscles during single-leg landings (Table 1). A study indicated that the amplitude of the medial gastrocnemius was significantly reduced when a brace was worn according to the results obtained via surface EMG [33]. We discovered that wearing a knee brace during single-leg landings significantly reduced not only the medial gastrocnemius force but also the lateral gastrocnemius force (Table 1 and Table 2). This could be a result of the maneuvers here being different to those studied by other researchers. A study discovered, using Opensim, that the peak soleus muscle force was much lower when a knee brace was worn [34], which is not exactly in line with our observations at the peak GRF. Our findings indicated that wearing a knee brace increased the soleus muscle force at 30 cm and decreased it at 45 cm (Table 1). The gastrocnemius muscle’s decreased activity as an ACL antagonist may be viewed as a protection mechanism against excessive anterior tibial translation [10,35]. Our findings reveal that wearing a knee brace may reduce quadricep muscle force, preventing possible ACL injuries. The soleus muscle has been shown to play an agonistic role in ACL loading [11]. Wearing a knee brace had opposite effects on the soleus at between the two heights. This suggests that wearing a knee brace may have different effects on the soleus muscle at different heights. Our findings indicate that wearing a knee brace causes changes in forces of muscles that do not cross the knee joint (soleus) during single-leg landings. As a result, we hypothesize that wearing a knee brace may contribute to ACL injury prevention by regulating gastrocnemius and soleus forces.

The data also confirmed our second hypothesis. Landing from a larger height increased the forces of the gluteus maximus, vastus medius, and vastus medius muscles by about 50% (Table 1 and Table 2). This was consistent with the finding of a significant increase in quadricep forces during a double-leg landing at 60 cm compared to 30 cm [36]. High knee muscle forces are usually sustained by soft tissue structures, including the ligaments and cartilage in the knee joint, especially at greater heights [37,38]. As a result, the risk of knee injury may increase as the landing height increases, and landing from greater heights should be avoided.

There are still several limitations associated with this study. Firstly, this study was conducted in a controlled laboratory environment and did not consider the effects of the external environment. Therefore, the landing tasks used in this study may not accurately represent landing maneuvers performed during training and competitions. Secondly, the human body is a complicated whole. While the single-leg landing procedure used here had been standardized and practiced, leg stance and speed were not precisely regulated in the same way during the landings. Thirdly, a more diverse selection of knee braces would have offered a more accurate indicator of the issue. Finally, EMG was not conducted in the study.

## 5. Conclusions

Our study demonstrated that bracing had a significant effect on the gluteus minimus, rectus femoris, vastus medialis, vastus lateralis, gastrocnemius medialis, gastrocnemius lateralis, soleus, and tibialis anterior muscles. Thus, wearing a brace may help prevent ACL injuries by regulating muscle forces during single-leg landings. The gluteus maximus, vastus medius, and vastus medius muscle forces were significantly increased in participants who were wearing a brace during the single-leg landings from 30 cm to 45 cm. This implied that single-leg landings from a greater height may increase the risk of knee joint injury. Further research is required to determine the effects of various braces on specific maneuvers.

## Figures and Tables

**Figure 1 ijerph-20-04652-f001:**
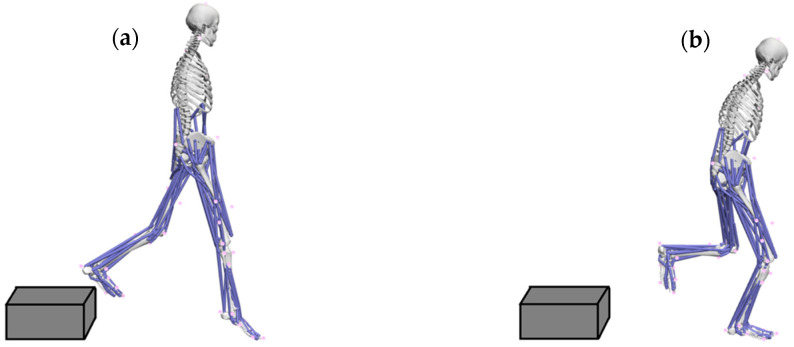
Key events of single-leg landing: (**a**) initial foot contact; (**b**) maximum knee flexion angle.

**Figure 2 ijerph-20-04652-f002:**
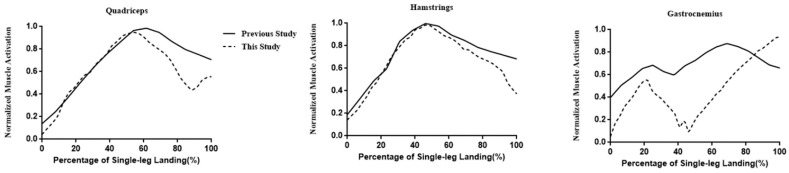
Comparisons between our simulation results and previous study data.

**Table 1 ijerph-20-04652-t001:** Muscle force data (N/kg) during landing of the braced and non-braced participants at peak GRF.

Muscle Force at Peak GRF	Height (cm)	Brace	No Brace	F Level B/N	F Level Height	F Level Interaction
Gluteus maximus	30	17.62 ± 7.04	19.60 ± 8.52	0.19	5.38 *	0.00
	45	17.14 ± 9.43	19.06 ± 8.72			
Gluteus medius	30	16.78 ± 9.16	19.04 ± 10.66	0.00	0.01	2.18
	45	19.12 ± 12.80	16.63 ± 9.41			
Gluteus minimus	30	3.29 ± 3.41	3.70 ± 4.56	0.08	1.42	2.66
	45	4.61 ± 5.40	2.76 ± 2.79			
Rectus femoris	30	1.65 ± 5.44	0.44 ± 0.98	3.83	1.38	0.22
	45	2.51 ± 5.37	1.95 ± 4.12			
Vastus medialis	30	8.87 ± 3.32	8.17 ± 3.26	112.22 *	0.28	0.23
	45	15.10 ± 5.03	15.02 ± 4.97			
Vastus intermedius	30	10.41 ± 3.75	9.65 ± 3.88	111.41 *	0.24	0.22
	45	17.79 ± 5.84	17.71 ± 5.76			
Vastus lateralis	30	19.10 ± 6.36	18.08 ± 7.17	106.19 *	0.25	0.09
	45	31.08 ± 9.35	30.72 ± 7.86			
Medial gastrocnemius	30	1.15 ± 1.90	1.16 ± 1.77	7.62 *	0.82	0.79
	45	3.20 ± 5.47	4.48 ± 4.48			
Lateral gastrocnemius	30	2.27 ± 3.38	3.4 ± 4.53	13.41 *	2.97	0.12
	45	5.84 ± 5.71	6.83 ± 6.31			
Soleus	30	32.09 ± 19.72	25.81 ± 20.27	25.65 *	0.95	3.17
	45	42.69 ± 19.30	44.01 ± 19.03			

* *p* < 0.05.

**Table 2 ijerph-20-04652-t002:** Muscle force data (N/kg) during landing of the braced and non-braced participants at maximum knee flexion.

Muscle Force at Maximum Knee Flexion	Height (cm)	Brace	No Brace	F Level B/N	F Level Height	F Level Interaction
Gluteus maximus	30	11.76 ± 5.50	12.22 ± 4.79	2.09	0.40	0.02
	45	12.93 ± 6.05	13.49 ± 5.54			
Gluteus medius	30	18.24 ± 7.80	20.59 ± 7.40	0.85	2.01	3.93
	45	20.49 ± 8.59	20.18 ± 8.94			
Gluteus minimus	30	4.48 ± 3.40	4.35 ± 2.93	4.66 *	2.91	1.80
	45	6.02 ± 4.54	4.97 ± 3.85			
Rectus femoris	30	2.98 ± 4.63	2.59 ± 4.08	10.53 *	1.23	0.20
	45	6.18 ± 6.73	5.20 ± 7.80			
Vastus medialis	30	10.43 ± 5.44	9.24 ± 5.05	103.61 *	4.97 *	0.00
	45	15.27 ± 3.77	14.08 ± 4.79			
Vastus intermedius	30	11.87 ± 5.82	10.60 ± 5.54	83.76 *	4.73 *	0.01
	45	16.89 ± 3.67	15.68 ± 4.84			
Vastus lateralis	30	19.56 ± 7.98	17.86 ± 8.20	44.74 *	2.73	0.39
	45	25.65 ± 4.28	24.82 ± 6.37			
Medial gastrocnemius	30	7.54 ± 9.14	8.99 ± 9.36	23.78 *	0.18	0.44
	45	16.07 ± 9.75	15.52 ± 9.47			
Lateral gastrocnemius	30	4.91 ± 4.41	4.35 ± 3.81	21.15 *	2.50	0.04
	45	7.50 ± 4.01	6.74 ± 3.87			
Soleus	30	45.27 ± 6.04	44.01 ± 11.17	0.95	0.47	0.08
	45	43.58 ± 9.90	43.10 ± 10.00			

* *p* < 0.05.

## Data Availability

The data used to support the findings of this study are available from the corresponding author upon request.

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
