# Peer review of "The Effect of a Knee Brace on Muscle Forces during Single-Leg Landings at Two Heights"

_ijerph, 2023, doi:10.3390/ijerph20054652_

Round 1
Reviewer 1 Report
In the paper 'The Effect of a Knee Brace on Muscle Forces During Single-Leg Landings at Two Heights', the authors studied the effect of a knee brace on lower limb muscles during certain movements. This reviewer recommends the publication of this manuscript after the authors answer the below questions.
1. Will the different weight of the participants have an effect on the simulation?
2. What is figure 1? Can the authors be more clear about the meaning of figure 1? Is it just drawing?
Author Response
"Please see the attachment."

Reviewer 2 Report
To authors
First, I would like to thank the authors for the opportunity to review their manuscript.
This study outlines the effects of knee braces in simple landings at two different heights (30 and 45 cm). Kinematics and Ground Reaction Forces were collected to estimate the activation of leg muscles using computational model software. Using a knee brace and the height affected the estimated muscle activity. Authors conclude that wearing knee braces can alter muscle activity in single leg landings. This knowledge is important as it conveys implications for the clinical setting. Results can also be useful in other areas of knowledge e.g., sports and physical exercise.
The study is generally well written, but the novelty is not clear enough. Please find my questions and concerns below.
Major concerns
Previous studies have shown that knee braces are effective in preventing knee injuries (please see refs [12,13]). The entire paper has been written around the assumption that altered muscle activation may be harmful for the knee joints. This assumption is not fundamentally correct and also it is not backed up by the current results. For example, several muscle activity estimations in the current study have shown to either increase or decrease when landing from different heights. It is yet not clear to the reader how is the increased/decreased muscle activation when landing associated to injuries. E.g., one would intuitively expect muscles to increase activity in order to compensate for the joint moments when landing. Please provide more information and possible interpretations to explain the change in the estimated muscle activity when wearing the knee braces.
Authors have used a computational software to estimate muscle activations/forces. There are two major limitations at this point:
- Authors state that the benefit of this technology is that deeper muscle activity can be included/estimated. However all but two muscles (gluteus minimus, vastus intermedius) reported in this study are superficial, hence their activity can be directly captured by surface EMG.
- To my knowledge, the software used to estimate muscle activities has been built to assess gait and has not been previously validated for single leg landing activities. Authors attempt to compare the estimated results with a previous study (figure 2), however providing no information regarding the data and the statistical measures used for this validation.
Presumably a more interesting analysis (subject to the availability of the data) would be to capture the actual EMG signal of -at least- the superficial knee muscles and validate the results against the computational model estimations.
Specific comments
Abstract, lines 41,42: The last sentence does not reveal a novel finding, is more than obvious. Also any association between wearing braces and knee injuries cannot be inferred from the current study design.
Introduction, lines 55,56: Authors may want to rework the aims and objectives of the study. The terms “biomechanical performance” and “affect lower limb muscles” seem too vague.
Methods, line 100: musculoskeletal muscle is redundant.
Methods line 119: medial femoris and lateral gastrocnemius are mentioned twice
Methods line 122: Why normalising with the body mass? Consider normalizing with baseline muscle activation. If no baseline activity has been captured, consider normalizing with the first second(s) of the trial (presumably there is enough data captured)
Results lines 136,137: The reason behind the inter-muscle comparison is unclear.
Author Response
"Please see the attachment."
